# The Prognostic Role of miR-31 in Head and Neck Squamous Cell Carcinoma: Systematic Review and Meta-Analysis with Trial Sequential Analysis

**DOI:** 10.3390/ijerph19095334

**Published:** 2022-04-27

**Authors:** Mario Dioguardi, Francesca Spirito, Diego Sovereto, Mario Alovisi, Riccardo Aiuto, Daniele Garcovich, Vito Crincoli, Luigi Laino, Angela Pia Cazzolla, Giorgia Apollonia Caloro, Michele Di Cosola, Andrea Ballini, Lorenzo Lo Muzio, Giuseppe Troiano

**Affiliations:** 1Department of Clinical and Experimental Medicine, University of Foggia, Via Rovelli 50, 71122 Foggia, Italy; spirito.francesca97@gmail.com (F.S.); diego_sovereto.546709@unifg.it (D.S.); elicio@inwind.it (A.P.C.); dott.dicosola@gmail.com (M.D.C.); lorenzo.lomuzio@unifg.it (L.L.M.); giuseppe.troiano@unifg.it (G.T.); 2Department of Surgical Sciences, Dental School, University of Turin, 10127 Turin, Italy; mario.alovisi@unito.it; 3Department of Biomedical, Surgical, and Dental Science, University of Milan, 20122 Milan, Italy; riccardo.aiuto@unimi.it; 4Department of Dentistry, Universidad Europea de Valencia, Paseo de la Alameda 7, 46010 Valencia, Spain; daniele.garcovich@universidadeuropea.es; 5Department of Basic Medical Sciences, Neurosciences and Sensory Organs, Division of Complex Operating Unit of Dentistry, “Aldo Moro” University of Bari, Piazza G. Cesare 11, 70124 Bari, Italy; vito.crincoli@uniba.it; 6Multidisciplinary Department of Medical-Surgical and Odontostomatological Specialties, University of Campania “Luigi Vanvitelli”, 80121 Naples, Italy; luigi.laino@unicampania.it; 7Unità Operativa Nefrologia e Dialisi, Presidio Ospedaliero Scorrano, ASL (Azienda Sanitaria Locale) Lecce, Via Giuseppina Delli Ponti, 73020 Scorrano, Italy; giorgiacaloro1983@hotmail.it; 8Department of Basic Medical Sciences, Neurosciences and Sensory Organs, University of Bari “Aldo Moro”, 70124 Bari, Italy; andrea.ballini@me.com; 9Department of Precision Medicine, University of Campania “Luigi Vanvitelli”, 80138 Naples, Italy

**Keywords:** miR-31, microRNA, OSCC, HNSCC, noncoding RNA, oral cancer

## Abstract

Background: Head and neck squamous cell carcinoma (HNSCC) is the sixth most common cancer worldwide with high recurrence, metastasis, and poor treatment outcome. Prognostic survival biomarkers can be a valid tool for assessing a patient’s life expectancy and directing therapy toward specific targets. Recent studies have reported microRNA (miR) might play a critical role in regulating different types of cancer. The main miR used as a diagnostic and prognostic biomarker and reported in the scientific literature for HNSCC is miR-21. Other miRs have been investigated to a lesser extent (miR-99a, miR-99b, miR-100, miR-143, miR-155, miR-7, miR-424, miR-183), but among these, the one that has attracted major interest is the miR-31. Methods: The systematic review was conducted following the PRISMA guidelines using electronic databases, such as PubMed, Scopus, and the Cochrane Central Register of Controlled Trials, with the use of combinations of keywords, such as miR-31 AND HNSCC, microRNA AND HNSCC, and miR-31. The meta-analysis was performed using the RevMan 5.41 software (Cochrane Collaboration, Copenhagen, Denmark). Results: This search produced 721 records, which, after the elimination of duplicates and the application of the inclusion and exclusion criteria, led to 4 articles. The meta-analysis was conducted by applying fixed-effects models, given the low rate of heterogeneity (*I*^2^ = 40%). The results of the meta-analysis report an aggregate hazard ratio (HR) for the overall survival (OS), between the highest and lowest miR-31 expression, of 1.59, with the relative intervals of confidence (1.22 2.07). Heterogeneity was evaluated through Chi^2^ = 5.04 df = 3 (*p* = 0.17) and the Higgins index *I*^2^ = 40; testing for the overall effect was Z = 3.44 (*p* = 0.00006). The forest plot shows us a worsening HR value of OS, in relation to the elevated expression of miR-31. Conclusions: In conclusion, the data resulting from the current meta-analysis suggest that miR-31 is associated with the prognosis of patients with HNSCC and that elevated miR-31 expression could predict a poor prognosis in patients with this type of neoplasm.

## 1. Introduction

Head and neck squamous cell carcinoma (HNSCC) is an aggressive life-threatening disease associated with high mortality rates [1]. The 5-year survival, despite the chemotherapy, radiotherapy, and surgery for stages 3 and 4 of the disease, is low (only 30% of the patients survive) [2].

The risk factors related to the onset are mainly tobacco and alcohol, but there is also a possible correlation with papilloma virus (HPV 16, HPV 18) [3]. Usually, HNSCC-HPV^+^ has a better prognosis compared to non-positive subjects, partly due to increased susceptibility to radiotherapy [4]. Therapy of patients with HNSCC, therefore, consists of radiotherapy (indicated in HPV^+^) in combination with chemotherapy and surgery, but potentially it involves an overall risk of disease recurrence of around 50% [5].

HNSCC are neoplasms, which, depending on the epithelium of origin, are recognized as laryngeal squamous cell carcinoma (LSCC), oropharyngeal squamous cell carcinoma (OPSCC), hypopharyngeal squamous cell carcinoma (HSCC), and oral squamous cell carcinoma (OSCC). This last one is the carcinoma with the highest incidence in the head and neck region, corresponding to around 60% of HNSCC [6]. The main cancer of the head and neck region is, however, represented by OSCC, which accounts for about 60% of all HNSCCs [6].

At the base of the etiopathogenetic processes of cancerization, we found genetic alterations that involve mutations of the main oncogenes (APC, p53, NF1, VHL, Rb BCL2, BRCA2, PTCH CD95, ST5, SWI/SNF, p16, YPEL3, ST7, and ST14) and oncogenes (Ras, raf, gsp, jun, fas, erbA, abl, sis erbB fms). In addition, events affecting cell cycle progression and proliferation, such as DNA methylation [7], histone modifications, and non-coding alterations of RNA [8] (such as microRNAs), can influence, cancer progression, stemness and resistance of cancer cells to therapeutics [9,10,11,12,13,14,15,16,17,18].

The microRNAs (miRs) are a large group of small single-stranded non-coding endogenous RNAs, approximately 18–25 nucleotides in length, which play a significant role in the post-transcriptional regulation of genes through the interaction with 3′UTR of target mRNA. The degradation or inhibition of their translation also plays an important role for cell development in differentiation, metabolism, proliferation, migration, induction of angiogenesis, and apoptosis, and they are stable molecules that can be found not only in tissues but also in body fluids, such as blood, saliva, and urine [19].

The miRs can express themselves in an altered way (downregulated or upregulated) in the different tumor forms. For HNSCC tumors, and more specifically for OSCC, the downregulated miRs mainly involved are: let-7a, let-7b, let-7c, let-7d, let-7e, let-7f, let-7g, let-7i, miR-26a, miR-99a-5p, miR-137, miR-139-5p, miR-143-3p, miR-184, and miR-375; while the upregulated ones are miR-21, miR-27a (-3p), miR-31, miR-93, miR-134, miR-146, miR-155, miR-196a, miR-196b, miR-211, miR-218, miR-222, miR-372, and miR-373 [20].

The different expression of miRs can be used as a prognostic biomarker, and the main miR established as a bioindicator, as described in the current scientific literature for HNSCC, is miR-21. Other miRs have been investigated to a lesser extent (miR-99a, miR-99b, miR-100, miR-143, miR-155, miR-7, miR-424, miR-183), but among these, the one that attracted the most interest is miR-31 [21,22,23].

Through the effect of RNA polymerase 2 intermediate forms of miR-31, pri-miR-31 and pre-miR-31 are assembled in succession, obtaining a pre-miR-31 of 70 nucleotides, which is transported out of the nucleus into the cytoplasm through exportin 5 (XPO5) and then cleaved by Dicer endonuclease in combination with TRBP, generating the miR-31 duplex complex (miRNA-31-3p and miRNA-31-5p) in the cytoplasm. Argonaute proteins (AGOs) play crucial roles in RNA-induced silencing complex (RISC) formation and activity. AGOs loaded with small RNA molecules (miRNA or siRNA) either catalyze the endoribonucleolytic cleavage of target RNAs or recruit factors responsible for translational silencing and target destabilization, forming a silencing complex while miR-31-5p degrades [24].

In the last 10 years, many studies have investigated the altered miR-31 expression, correlating it to the prognosis and progression of tumor growth [25,26,27,28]. A clinical study on 56 patients conducted by Qiang et al. (2019) [29] investigated the prognostic potential by identifying a longer survival period in patients presenting a lower expression of miR-31, indicating how this non-coding RNA sequence can be considered a crucial reference indicator for the prognosis of patients with HNSCC. These data are partially in agreement with the results of Tu et al. (2021) [30], which identify a worse overall survival in patients with a higher expression of miR-31. These two very recent studies highlight how miR-31, as well miR-21, could represent a prognostic biomarker of survival. Furthermore, no other systematic reviews have been conducted on the prognostic role of the altered expression of miR-31. Taking in account these premises, the aim of this systematic review and meta-analysis was to investigate the prognostic potential of miR-31 as a survival biomarker for HNSCC in the light of new miR-31 studies, providing a pooled hazard ratio (HR) value (high and low expression of miR-31) on the overall survival (OS) in patients presenting HNSCC.

## 2. Materials and Methods

### 2.1. Protocol

The drafting of the review was carried out following the indications of the Preferred Reporting Items for Systematic Reviews and Meta-Analysis (PRISMA) [31]. The protocol with which the systematic review was performed was established before proceeding with the search and screening of records in data banks.

### 2.2. Eligibility Criteria

All prospective and retrospective studies and randomized controlled trials (RCTs) that evaluated differences in the tissue expression of miR-31 in HNSCC in correlation with prognostic survival indices were considered, with the following PICO question: Participants (patients with HNSCC), Intervention (altered expression of miR-31 in HNSCC), Control (patients with HNSCC who have low expression of miR-31), Outcome (difference in prognosis of survival among patients with low and high miR-31 expression in HNSCC). The PICO question, therefore, was as follows: Is there a difference in OS between HNSCC patients with high miR-31 expression versus those with low expression?

The inclusion criteria were as follows: all study reports that indicated and reported data on prognostic survival indices (Kaplan–Meier curves, hazard ratio, Cox regression) between low and high miR-31 expression in tumor tissues (HNSCC) were included.

The exclusion criteria were as follows: studies published in a language other than English, those that did not present data on survival and the expression of miR-31, and those at high risk of bias.

### 2.3. Sources of Information, Research, and Selection

The studies were identified through a literature search in electronic databases by two researchers (M.D. and D.S.). Limits relating to the language of publication were applied, and articles in a language other than English were excluded. The literature search was conducted on the PubMed, Scopus, and Cochrane databases. The last bibliographic search was conducted on 3 March 2022, for a further update of the Bibliography. In addition, a search of the gray literature on Google Scholar was also conducted, and the bibliographic sources of previous systematic reviews on the miR-31 in HNSCC were investigated.

We used the following terms to search the databases: miR-31 AND HNSCC, Microrna AND HNSCC AND prognosis, miR-31 AND OSCC, miR-31 AND laryngeal cancer. Duplicates were removed using EndNote and manually. The articles identified were independently evaluated and scrutinized by two reviewers (M.D. and D.S.) for title and abstract analysis and for potentially eligible studies and text analyses for inclusion in the systematic review. Furthermore, the k agreement between the three reviewers was assessed. A third reviewer (A.B.) had the task of resolving the situations of disagreement.

### 2.4. Data Collection Process, Data Characteristics

The data to be extracted were previously established by the two reviewers (M.D. and D.S.) responsible for screening the studies and were reported in two tables independently to be subsequently compared to reduce the risk of errors. The data that were extracted from the articles concerned the type of study, the year of publication, the first author, the country that conducted the study, the number of patients, the type of HNSCC, the type of miR investigated, the cut-off between low and high expression, hazard ratio (HR), and the presence of a Kaplan–Meier curve. Specifically, all the HR data between high expression vs. low expression of miR-31 in tumor tissues (HNSCC) concerning the OS were searched and extracted, and secondly, data concerning disease-free survival (DFS), recurrence-free survival (RFS), progression-free survival (PFS), and cancer-specific survival (CSS) were searched and extracted if the data were in the form of Kaplan–Meier curves. The HR data were obtained through the method of Tierney et al., using a point and curve acquisition software (Engauge digitizer) and subsequently using an Excel spreadsheet, dedicated and available as additional material in the publication by Tierney et al. [32].

### 2.5. Risk of Bias in Individual Studies, Summary Measures, Summary of Results, Risk of Bias between Studies, Additional Measures

The risk of bias in the individual studies was assessed by an author (M.D.), with a second author tasked with verifying the correct assessment (D.S.). Parameters derived from the Reporting Recommendations for prognostic studies of markers (REMARK) were used for the assessment, and studies with a high risk of bias were excluded from the meta-analysis. The results were represented by forest plots, and inconsistency indices, such as the Higgins index *I*^2^, were evaluated.

The risk of bias between the studies was assessed graphically through the analysis of the overlaps of the confidence intervals, through the *I*^2^ inconsistency index (an *I*^2^ value greater than 75% was considered high, and a random effects analysis was applied in specific cases), and through a funnel plot. In the presence of a meta-analysis with high heterogeneity indices, a sensitivity analysis could be performed, excluding only the studies that presented a low overlap of the confidence intervals or that emerged graphically from the funnel plot. For the meta-analysis, and in particular for the calculation of the pooled HR, the software Reviewer Manager 5.4 (Cochrane Collaboration, Copenhagen, Denmark) was used. We used the GRADE pro-Guideline Development Tool online software (GRADEpro GDT, Evidence Prime, Hamilton, ON, Canada) to assess the quality of the evidence.

The trial sequential analysis (TSA) was performed using Stata 13 (StataCorp, College Station, TX, USA), with the implementation of the R 4.2 software (Foundation for Statistical Computing, Vienna, Austria) and by installing the idbounds and metacumbounds commands.

## 3. Results

### 3.1. Selection of Studies

The search in the Scopus, PubMed, and Cochrane Central Register of Controlled Trial databases provided 716 bibliographic citations. Following the removal of overlaps, 512 registrations were obtained, of which 459 were excluded because it was found that they did not meet the eligibility criteria upon reading the abstracts, and 15 articles were potentially admissible but only 4 met the inclusion and exclusion criteria and were included in the meta-analysis. Furthermore, the analysis of the gray literature (Google Scholar, Open gray) and previous systematic reviews did not allow the identification of further studies to be included in the meta-analysis (Figure 1). Finally, an update of the research on the sources was carried out on 3 March 2022, with the addition of records from Web of Science and with the updating of keywords and results on Scopus and PubMed. All keywords and records search details are also represented in Table 1.

### 3.2. Data Characteristics

Articles included in the meta-analysis are as follows: Jakob et al. (2019) [33], Wang et al. (2018) [34], Tu et al. (2021) [30], and Qiang et al. (2019) [29]. In total, three studies were excluded, despite presenting HR values for OS and PFS, because they did not fully meet the eligibility criteria. In the studies of Gao et al. (2013) [35] and Chen et al. (2018) [36], the HR values referred to a microRNA expression signature composed of five miR, including miR-31; while for the study of Hung et al. (2016) [36], the data on PFS referred to potentially malignant disorders.

The characteristics of the extracted data are described in the Materials and Methods section for the survival data expressed in the form of a Kaplan–Meier curve and HR and extracted according to the Tierney method [32]. All extracted data are reported in Table 2.

Only four articles were included in the meta-analysis. All four studies included are retrospective clinical studies whose results were published between 2018 and 2021 and included a follow-up period of 5 to 13 years (Tu et al. [30]). The total number of HNSCC patients recruited is 240.

Studies by Tu et al. (2021) [30] and Jakob et al. (2019) [33] investigated patients with OSCC; Qiang et al. (2019) [29] considered HSCC and LSCC, while Wang et al. (2018) [34] considered more generally the HNSCC without specifying the site of the neoplasm.

All four studies considered OS as a prognostic parameter; Jakob (2019) [33] also used DFS and RFS, while Wang et al. used only DFS [34]. All the data concerning the prognostic survival indices were extracted and reported in Table 3.

### 3.3. Risk of Bias in Studies

The risk of bias was assessed through parameters derived from the REMARK. Based on the REMARK guidelines, a score from 0 to 3 was considered for each factor.

### 3.4. Meta-Analysis

The meta-analysis was conducted by applying fixed-effects models, given the low rate of heterogeneity (*I*^2^ = 38%). The results of the meta-analysis report aggregate HR for OS. Between high and low miR-31 expression of 1.58, with the relative intervals of confidence [1.21, 2.06], heterogeneity was evaluated through Chi^2^ = 4.80 df = 3 (*p* = 0.19) and the Higgins index *I*^2^ = 38. Testing for the overall effect was Z = 3.39 (*p* = 0.0007). The forest plot presents the black diamond in a position of worsening of OS, in relation to the high miR-31 expression (Figure 2).

### 3.5. Risk of Bias across Study

The risk of bias between the studies is low (*I*^2^ = 38%), as evidenced by the overlapping of the confidence intervals, and further confirmation comes from the funnel plot, which does not identify sources of heterogeneity (Figure 3).

To reduce the weight of the study by Qiang et al. of 76.1% in the meta-analysis, it was decided to also apply a random-effects model. In fact, the weight of the study on the meta-analysis drops to 50% with an HR value of 1.86 [1.18, 2.93], always in favor of the worsening of OS (Figure 4).

### 3.6. Trial Sequential Analysis, Grade

Trial sequential analysis (TSA) was performed to evaluate the power of the meta-analysis, adjusting the results to avoid type I and II errors. The program used was Stata 13 (StataCorp, College Station, TX, USA), with the integration of the R 4.2 software (Foundation for Statistical Computing, Vienna, Austria) through the Metacumbounds commands, as described by Miladinovic et al. [38]. The O’Brien—Fleming spending function was used by applying random effects. The accrued information size (AIS) and, subsequently, a priori information size (APIS) commands were used by the Dialog BOX to determine the optimal sample size and assuming a reduction risk relative (RRR) of 38%, an Alpha value equal to 5% (type 1 error), and beta at 20% (type 2 error) (Figure 5) for the power of the results.

The TSA curve crosses the line Z = 1.98, and crossing of the monitoring boundary before reaching the information size provides for firm evidence of effect. The APIS graph shows that, for an RRR of 38%, alpha 5%, and for a power of 80%, the number of optimal patients is 571.

The authors also used GRADE pro-GDT to assess the quality of the primary outcome (Table 4). The results suggested that the quality of evidence is low for outcome as the primary result.

## 4. Discussion

We conducted a systematic review through a meta-analysis of the data published in the literature on HNSCC to establish and expose a complete picture of the role of miR-31 as a prognostic biomarker of the altered expression of tissue miR-31 in HNSCC. To our knowledge, this is the very first meta-analysis review describing the role of miR-31 in HNSCC prognosis, in which four studies were analyzed with a total of 240 patients.

Many studies confirm that miR-31 is found to be upregulated under HNSCC, including a recent systematic review conducted by Al Rawi et al. 2021 [39] on the salivary expression of non-coding RNA.

In fact, Al-Rawi reports miR-31 among the 12 upregulated microRNAs during HNSCC and among the 4 non-coding RNAs investigated in relation to squamous cell carcinomas of the head and neck region [39]. On the one hand, the prognostic power of miR-31 tissue expression has been investigated; on the other hand, only few recent studies and literature data are in partial agreement with different degrees of significance [28,29,32,33].

Wang et al. (2018) [34] demonstrated that miR-31 tissue levels were increased in HNSCC patients and closely associated with aggressive clinical variables. Furthermore, the elevated expression of miR-31 in tissues was positively correlated with shorter OS (HR 3.31 *p* = 0.015) and DFS, demonstrating that miR-31 tissue expression is an independent predictor for both OS and DFS of HNSCC and indicating how miR-31 could serve as a prognostic marker for HNSCC.

Jakob et al. (2019) [33] observed an over-regulated expression level of hsa-mir-31-5p in OSCC but without finding a correlation with clinical–pathological characteristics but associating it with a poorer OS (HR 3.69 p 0.028), as well as for the PFS.

Jakob et al. and Wang et al. report similar HR values (3.69 and 3.31), as also shown by the forest plot (Figure 2) and Table 2 [33,34].

Qiang et al. (2019) indicates, in agreement with Jakob and Wang [30,31], that the expression of miR-31 can be considered a crucial prognostic biomarker for the prognosis of patients with HNSCC, correlating it also with the clinical–pathological characteristics of the patient [29].

Analyzing and extrapolating the HR data from the Kaplan–Meier curve from the study by Tu et al. (2021) [30] (HR 1.68, C.I. = 0.77–3.64 *p* = 0.189) reveals a trend in which patients with higher miR-31 expression had worse OS but with differences that were not statistically significant.

Studies conducted on the Cancer Genome Atlas (TCGA) database report data that are not statistically significant; in fact, Jakob et al. identify an OS HR of 0.64 *p* = 0.19 for the TCGA cohort [33]. Furthermore, miR-31 has been included in prognostic predictor models with other miRs (miR-99a, miR-31, miR-410, miR-424, and miR-495) to evaluate the survival response to radiotherapy [36].

Our meta-analysis reports data that are statistically significant. In fact, aggregate HR for OS is 1.58 C.I. [1.21 2.06], Chi^2^ = 4.80, df = 3 (*p* = 0.19), giving useful information on how the altered expression of tissue miR-31 may be a valid prognostic biomarker. The data of our meta-analysis are in line with the four included studies, of which only three report statistically significant data. Indeed, by analyzing the confidence interval of HR in the forest plot (Figure 2), we can see how Tu et al. (2021) [30] intersect the center line to no effect, while Qiang (2019) comes very close to it [29]. The risk of bias within the studies is acceptable for all four studies, excluding only Tu et al. (2021) [30]. This last study has a slightly lower score, determined by the fact that the data (HR) for the outcome investigated in this meta-analysis were extrapolated by the authors of this review, starting from the Kaplan–Meier curve not being represented in a numerical value, and not being provided in the form of supplementary material. The risk of bias between the studies was low, as evidenced by the funnel plot and the heterogeneity indices (*I*^2^ = 38%).

Our systematic review through the meta-analysis shows some limitations. The first limitation is represented by a limited number of data and studies on the prognostic role of miR-31 expression in HNSCC, with only 240 patients included. In fact, further clinical studies are needed to evaluate the association between impaired expression of miR-31 and OS. The second limit is represented by the fact that the data reported by Tu et al. [30] on the HR were in the form of a Kaplan–Meier curve, and the data were extrapolated from the curve using the Tierney method [32]. This method, although widely validated, is not free from errors and could provide a slightly different HR value from the real one. This limit also affects the quality of evidence provided by the GRADE (Table 3).

Another limitation comes from the weight of Qiang et al. study [29], which presents an overweight of 76.1% in the meta-analysis, applying fixed effects—a limit that can be partially overcome by applying random effects. Certainly, the weight of the study on the meta-analysis drops to 50%, and in any case, the pooled HR is 1.86 [1.18, 2.93], always in favor of the worsening of OS.

Finally, the data from the current meta-analysis suggest that miR-31 is associated with the prognosis of patients with HNSCC and that high miR-31 expression could predict a poor prognosis in patients with this type of malignancy. From the HR, sample number, and standard error data extracted from the meta-analysis, the TSA was also conducted. The data results indicate that there is statistical power in the data, although from the APIS graphs (Figure 5) with a RRR of 38%, the ideal total number of people to have a power of both 80% is 571 patients. In our meta-analysis, the total number of patients included was 240. So, even if in the presence of a valid statistical power, it is not wrong to say that further studies on the issue are needed.

## 5. Conclusions

In conclusion, the data from the meta-analysis suggest that elevated miR-31 expression is associated with poor prognosis in patients with HNSCC. Consequently, exhaustive investigations of miRNA, for instance regarding intercommunication among miRNAs and between miRNAs and other genes, altered protein expression induced by miRNAs, and site-specific miRNA expression profiling, are, therefore, prerequisites before future clinical trials of therapeutic applications can go ahead.

## Figures and Tables

**Figure 1 ijerph-19-05334-f001:**
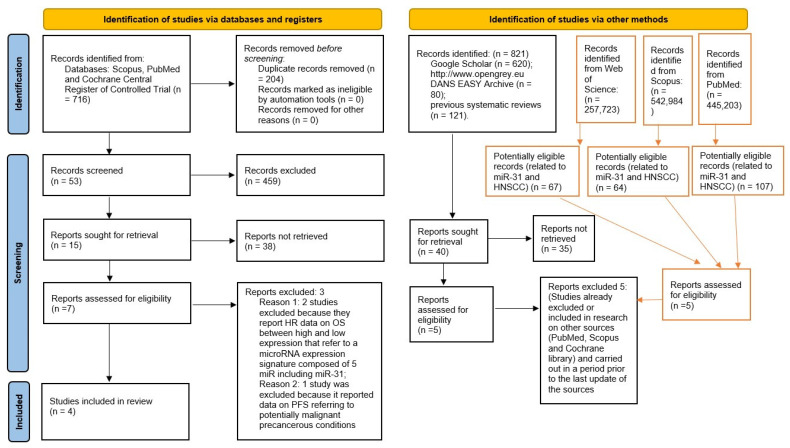
Entire selection and screening procedures are described in the PRISMA flowchart; tables with the red lines are the searches performed subsequently (on 3 March 2022), with the addition of Web of Science and with an update of the keywords and results on PubMed and Scopus.

**Figure 2 ijerph-19-05334-f002:**
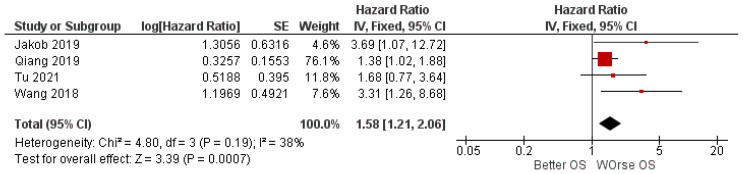
Forest plot of the fixed-effects model of the meta-analysis; HR = 1.58, 95% CI: [1.21, 2.06]; df = degrees of freedom; *I*^2^ = Higgins heterogeneity index, *I*^2^ < 50%, heterogeneity irrelevant; *I*^2^ > 75%, significant heterogeneity; C.I. = confidence intervals; P = *p* value; SE = standard error. The graph for each study shows the first author and the date of publication, hazard ratio with confidence intervals, log HR standard error, and weight of each study expressed as a percentage. The final value is expressed in bold with the relative confidence intervals. The black line shows the position of the average value, and the rhombus in light black shows the measure of the average effect.

**Figure 3 ijerph-19-05334-f003:**
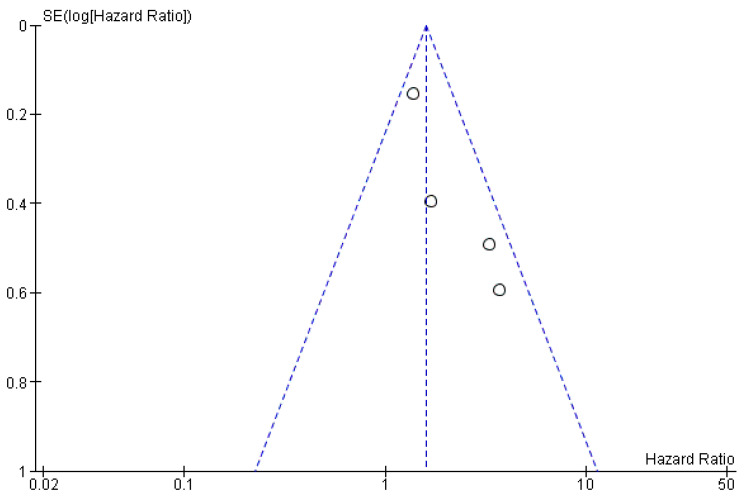
Funnel plot. The absence of heterogeneity is evident graphically. *I*^2^ = 38%, SE: standard error.

**Figure 4 ijerph-19-05334-f004:**
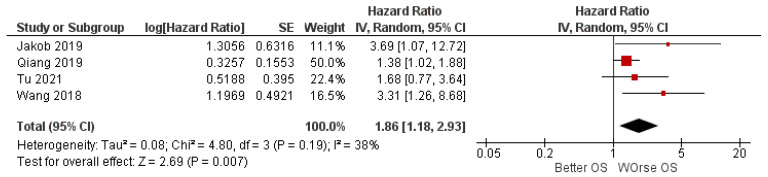
Forest plot of the random-effects model of the meta-analysis.

**Figure 5 ijerph-19-05334-f005:**
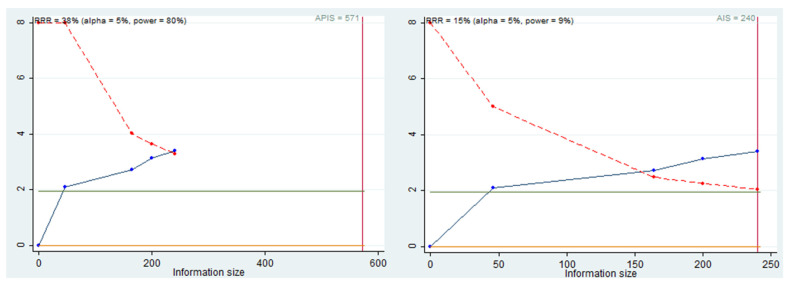
AIS, APIS, light green line (Z = 1.98), dashed red line (monitoring boundary), blue line (cumulative z curve), red line (sample size).

**Table 1 ijerph-19-05334-t001:** Complete overview of the search methodology. In total, 716 records were identified by the databases (512 records after removal of duplicates); Records identified by other research sources: 721 records.

Identification Records through Databases and Registers			
Databases	k-Words	Search Details	Number
PubMed	miR-31 AND HNSCC	Search: miR-31 AND HNSCC Sort by: Most Recent “miR-31”[All Fields] AND (“hnsccs”[All Fields] OR “squamous cell carcinoma of head and neck”[MeSH Terms] OR (“squamous”[All Fields] AND “cell”[All Fields] AND “carcinoma”[All Fields] AND “head”[All Fields] AND “neck”[All Fields]) OR “squamous cell carcinoma of head and neck”[All Fields] OR “hnscc”[All Fields]) Translations HNSCC: “hnsccs”[All Fields] OR “squamous cell carcinoma of head and neck”[MeSH Terms] OR (“squamous”[All Fields] AND “cell”[All Fields] AND “carcinoma”[All Fields] AND “head”[All Fields] AND “neck”[All Fields]) OR “squamous cell carcinoma of head and neck”[All Fields] OR “hnscc”[All Fields]	34
	Microrna AND HNSCC AND prognosis	Search: Microrna AND HNSCC AND prognosis Sort by: Most Recent (“microrna s”[All Fields] OR “micrornas”[MeSH Terms] OR “micrornas”[All Fields] OR “microrna”[All Fields]) AND (“hnsccs”[All Fields] OR “squamous cell carcinoma of head and neck”[MeSH Terms] OR (“squamous”[All Fields] AND “cell”[All Fields] AND “carcinoma”[All Fields] AND “head”[All Fields] AND “neck”[All Fields]) OR “squamous cell carcinoma of head and neck”[All Fields] OR “hnscc”[All Fields]) AND (“prognosis”[MeSH Terms] OR “prognosis”[All Fields] OR “prognoses”[All Fields]) Translations Microrna: “microrna’s”[All Fields] OR “micrornas”[MeSH Terms] OR “micrornas”[All Fields] OR “microrna”[All Fields] HNSCC: “hnsccs”[All Fields] OR “squamous cell carcinoma of head and neck”[MeSH Terms] OR (“squamous”[All Fields] AND “cell”[All Fields] AND “carcinoma”[All Fields] AND “head”[All Fields] AND “neck”[All Fields]) OR “squamous cell carcinoma of head and neck”[All Fields] OR “hnscc”[All Fields] prognosis: “prognosis”[MeSH Terms] OR “prognosis”[All Fields] OR “prognoses”[All Fields]	395
	miR-31 AND OSCC	Search: miR-31 AND OSCC Sort by: Most Recent “miR-31”[All Fields] AND “OSCC”[All Fields]	25
	miR-31 AND laryngeal cancer	Search: miR-31 AND laryngeal cancer Sort by: Most Recent “miR-31”[All Fields] AND (“laryngeal neoplasms”[MeSH Terms] OR (“laryngeal”[All Fields] AND “neoplasms”[All Fields]) OR “laryngeal neoplasms”[All Fields] OR (“laryngeal”[All Fields] AND “cancer”[All Fields]) OR “laryngeal cancer”[All Fields]) Translations laryngeal cancer: “laryngeal neoplasms”[MeSH Terms] OR (“laryngeal”[All Fields] AND “neoplasms”[All Fields]) OR “laryngeal neoplasms”[All Fields] OR (“laryngeal”[All Fields] AND “cancer”[All Fields]) OR “laryngeal cancer”[All Fields]	7
	miR-31 AND oropharynx	Search: miR-31 AND oropharynx Sort by: Most Recent “miR-31”[All Fields] AND (“oropharynx”[MeSH Terms] OR “oropharynx”[All Fields] OR “oropharynxes”[All Fields]) Translations oropharynx: “oropharynx”[MeSH Terms] OR “oropharynx”[All Fields] OR “oropharynxes”[All Fields]	1
Total PubMed			462
Duplicates removed (PubMed) EndNote			432
SCOPUS	miR-31 AND HNSCC	TITLE-ABS-KEY (mir-31 AND hnscc)	13
	miR-31 AND OSCC	TITLE-ABS-KEY (mir-31 AND oscc)	27
	Microrna AND HNSCC AND prognosis	TITLE-ABS-KEY (microrna AND hnscc AND prognosis)	211
Total SCOPUS			251
Duplicates removed (PubMed and SCOPUS) EndNote			535
Cochrane library	miR-31 AND HNSCC	miR-31 AND HNSCC in Title Abstract Keyword	1
	Microrna AND HNSCC AND prognosis	Microrna AND HNSCC AND prognosis in Title Abstract Keyword	2
Total Cochrane library			3
Total records			716
Number of records after duplicates removed (PubMed Scopus and Cochrane library) EndNote			536
Number of records after duplicates removed MANUAL			512
Other research sources, gray literature, and previous systematic reviews			
Google Scholar	Mir 31	allintitle: “mir 31”	620
http://www.opengrey.eu (accessed on 5 January 2022) DANS EASY Archive	Mir		80
Previous systematic reviews			121
Update of the research completed on 3 March 2022 with the inclusion of records from Web of Science			
Web of Science		Head and Neck Cancer	81,593
		Oral cancer	104,022
		Oral carcinoma	47,461
		Oral squamous cell carcinoma	28,540
		Head and Neck Squamous Cell Carcinoma	38,318
		Pharyngeal cancer	3766
		miR-31	972
Total Web Of Science			257,723
PubMed	Oral carcinoma OR Oral Cancer OR Head and Neck Cancer OR Oral squamous cell carcinoma OR Head and Neck Squamous Cell Carcinoma OR pharyngeal cancer	Search: Oral carcinoma OR Oral Cancer OR Head and Neck Cancer OR Oral squamous cell carcinoma OR Head and Neck Squamous Cell Carcinoma OR pharyngeal cancer ((“mouth”[MeSH Terms] OR “mouth”[All Fields] OR “oral”[All Fields]) AND (“carcinoma”[MeSH Terms] OR “carcinoma”[All Fields] OR “carcinomas”[All Fields] OR “carcinoma s”[All Fields])) OR (“mouth neoplasms”[MeSH Terms] OR (“mouth”[All Fields] AND “neoplasms”[All Fields]) OR “mouth neoplasms”[All Fields] OR (“oral”[All Fields] AND “cancer”[All Fields]) OR “oral cancer”[All Fields]) OR (“head and neck neoplasms”[MeSH Terms] OR (“head”[All Fields] AND “neck”[All Fields] AND “neoplasms”[All Fields]) OR “head and neck neoplasms”[All Fields] OR (“head”[All Fields] AND “neck”[All Fields] AND “cancer”[All Fields]) OR “head and neck cancer”[All Fields]) OR (“squamous cell carcinoma of head and neck”[MeSH Terms] OR (“squamous”[All Fields] AND “cell”[All Fields] AND “carcinoma”[All Fields] AND “head”[All Fields] AND “neck”[All Fields]) OR “squamous cell carcinoma of head and neck”[All Fields] OR (“oral”[All Fields] AND “squamous”[All Fields] AND “cell”[All Fields] AND “carcinoma”[All Fields]) OR “oral squamous cell carcinoma”[All Fields]) OR (“squamous cell carcinoma of head and neck”[MeSH Terms] OR (“squamous”[All Fields] AND “cell”[All Fields] AND “carcinoma”[All Fields] AND “head”[All Fields] AND “neck”[All Fields]) OR “squamous cell carcinoma of head and neck”[All Fields] OR (“head”[All Fields] AND “neck”[All Fields] AND “squamous”[All Fields] AND “cell”[All Fields] AND “carcinoma”[All Fields]) OR “head and neck squamous cell carcinoma”[All Fields]) OR (“pharyngeal neoplasms”[MeSH Terms] OR (“pharyngeal”[All Fields] AND “neoplasms”[All Fields]) OR “pharyngeal neoplasms”[All Fields] OR (“pharyngeal”[All Fields] AND “cancer”[All Fields]) OR “pharyngeal cancer”[All Fields]) Translations Oral: “mouth”[MeSH Terms] OR “mouth”[All Fields] OR “oral”[All Fields] carcinoma: “carcinoma”[MeSH Terms] OR “carcinoma”[All Fields] OR “carcinomas”[All Fields] OR “carcinoma’s”[All Fields] Oral Cancer: “mouth neoplasms”[MeSH Terms] OR (“mouth”[All Fields] AND “neoplasms”[All Fields]) OR “mouth neoplasms”[All Fields] OR (“oral”[All Fields] AND “cancer”[All Fields]) OR “oral cancer”[All Fields] Head and Neck Cancer: “head and neck neoplasms”[MeSH Terms] OR (“head”[All Fields] AND “neck”[All Fields] AND “neoplasms”[All Fields]) OR “head and neck neoplasms”[All Fields] OR (“head”[All Fields] AND “neck”[All Fields] AND “cancer”[All Fields]) OR “head and neck cancer”[All Fields] Oral squamous cell carcinoma: “squamous cell carcinoma of head and neck”[MeSH Terms] OR (“squamous”[All Fields] AND “cell”[All Fields] AND “carcinoma”[All Fields] AND “head”[All Fields] AND “neck”[All Fields]) OR “squamous cell carcinoma of head and neck”[All Fields] OR (“oral”[All Fields] AND “squamous”[All Fields] AND “cell”[All Fields] AND “carcinoma”[All Fields]) OR “oral squamous cell carcinoma”[All Fields] Head and Neck Squamous Cell Carcinoma: “squamous cell carcinoma of head and neck”[MeSH Terms] OR (“squamous”[All Fields] AND “cell”[All Fields] AND “carcinoma”[All Fields] AND “head”[All Fields] AND “neck”[All Fields]) OR “squamous cell carcinoma of head and neck”[All Fields] OR (“head”[All Fields] AND “neck”[All Fields] AND “squamous”[All Fields] AND “cell”[All Fields] AND “carcinoma”[All Fields]) OR “head and neck squamous cell carcinoma”[All Fields] pharyngeal cancer: “pharyngeal neoplasms”[MeSH Terms] OR (“pharyngeal”[All Fields] AND “neoplasms”[All Fields]) OR “pharyngeal neoplasms”[All Fields] OR (“pharyngeal”[All Fields] AND “cancer”[All Fields]) OR “pharyngeal cancer”[All Fields]	445,203
Scopus		TITLE-ABS-KEY (“Oral carcinoma” OR “Oral Cancer” OR “Head and Neck Cancer” OR “Oral squamous cell carcinoma” OR “Head and Neck Squamous Cell Carcinoma” OR “pharyngeal cancer”)	97,781
Total of updated records			542,984

**Table 2 ijerph-19-05334-t002:** The data extracted from the four articles included in the meta-analysis and from the three articles excluded are reported in the present table. R (radiotherapy), OPMD (oral potentially malignant disorder).

First Author, Data	Country	Study Design	Number of Patients	Follow-Up Max	Tumor Type\Tumor Site	Cut-off	miR	HR miR-31 Low and High Expression (OS, PFS, CSS, DFS, RFS)
Jakob (2019) [33]	Germany	RT	36	60 months	OSCC	median	miR-21, miR-29, miR-31, miR-99a, miR-99b, miR-100, miR-143, miR-155.	OS:HR 3.69 (1.07–12.79) *p* = 0.028 RFS:HR 1.82(0.66–5.05) *p* = 2.4297 PFS: HR 2.31 (0.94–5.69) *p* = 0.05982
Wang (2018) [34]	China	RT	118	60 months	HNSCC	median	miR-31	OS: HR 3.31 (1.42–5.36) * *p* = 0.015 DFS: HR 3.86 (1.53–6.05) *p* = 0.009
Qiang (2019) [29]	China	RT	46	60 months	21 HSCC, 25 LSCC	median	miR-31	OS: HR 1.38 (1.02–1.87) *p* = 0.036
Tu (2021) [30]	Taiwan	RT	40	160 months	OSCC	median	miR-31	OS: HR 1.68 (0.7747–3.6433) *p* = 0.189
Chen (2018) [36] ^1^	China	RT	509 (307 con R)	80 months	HNSCC	median	miR-99a, miR-31 miR-410, miR-424, miR-495	OS R: HR 3.65, (2.46–8.16) *p* < 0.0001; OS: HR 1.81 (1.45–2.57) *p* < 0.0001
Gao (2013) [35] ^2^	USA	RT	150	50 months	OPSCC	median	miR-9, miR-223, miR-31, miR-18a, miR-155	OS HR 3.22 *p* = 0.0022
Hung (2016) [37] ^3^	Taiwan	Prospective	46	28 months	OPMD	median	miR-21, miR-31	PFS:HR 8.43 (1.04–68.03) *p* = 0.047

* There is probably an error in reporting the confidence interval values in the study by Wang et al. For *p* values of 0.015 and HR of 3.31, these confidence intervals seem not to be possible. In fact, ReV Manager 5.4 reports, for the confidence interval, a value of (1.42 5.36) and an HR of 2, 7. For the determination and implementation of the confidence intervals in the ReV manager 5.4 software (Cochrane Collaboration, Copenhagen, Denmark) we proceeded using the *p* value of 0.15 and HR 3.31. ^1^ The study was not included in the meta-analysis because the HR value refers to the OS of HNSCC between high and low expressions overall of five miR, including miR-31. Furthermore, the data derive from the TGCA. ^2^ The study was not included in the meta-analysis because the HR value refers to the OS of OPSCC between high and low overall expression of five miR (miR-9, miR-223, miR-31, miR-18a, miR-155). ^3^ The study was excluded because it reported data on PFS referring to potentially malignant precancerous conditions.

**Table 3 ijerph-19-05334-t003:** Assessment of risk of bias within the studies, with scores 8 to 10 = low quality, 11 to 14 = intermediate quality, and 15 to 18 = high quality.

First Author, Data	Sample	Clinical Data	Marker Quantification	Prognostication	Statistics	Classical Prognostic Factors	Score
Wang (2018) [34]	3	2	3	2	2	3	15
Qiang (2019) [29]	2	3	3	2	2	2	14
Tu (2021) [30]	1	3	3	2	2	2	13
Jakob (2019) [33]	1	3	3	3	3	3	16

**Table 4 ijerph-19-05334-t004:** Evaluation of GRADE pro-GDT.

Certainty Assessment							No. of Patients	Effect		Certainty
No. of Studies	Study Design	Risk of Bias	Inconsistency	Indirectness	Imprecision	Other Considerations		Relative (95% CI)	Absolute (95% CI)	
4	observational studies	not serious	not serious	not serious	serious ^1^	strong association	240	HR 1.58 (1.21 to 2.06)	2 fewer per 1.000 (From 2 fewer to 1 fewer)	⊕⊕◯◯ Low

^1^ In Tu et al., the HR value was extrapolated from the Kaplan–Meier curves reported in the manuscripts. This passage is not free from errors and can be a source of inaccuracy. CI, confidence interval; HR, hazard ratio.

## Data Availability

Data are contained within the article.

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
