# Peer review of "The Prognostic Role of miR-31 in Head and Neck Squamous Cell Carcinoma: Systematic Review and Meta-Analysis with Trial Sequential Analysis"

_ijerph, 2022, doi:10.3390/ijerph19095334_

Round 1
Reviewer 1 Report
The revised manuscript has responded well to the initial review. However, on lines 284 and 382, the fixed-effects weight attributed to Qiang 2019 was 78%, whereas Figure 2 gives it as 76%.This should be corrected. Also, on line 296, "Fleminge" should be changed to "Fleming".
Author Response
reviewer
The revised manuscript has responded well to the initial review. However, on lines 284 and 382, the fixed-effects weight attributed to Qiang 2019 was 78%, whereas Figure 2 gives it as 76%. This should be corrected. Also, on line 296, "Fleminge" should be changed to "Fleming".
Answer
We thank the Reviewers for the considerable attention and the valuable comments that certainly helped us to improve the quality of the present paper. We have revised the manuscript according to the Reviewers’ comments.).
- all errors have been corrected.
Please let us know if the revised paper satisfies requirements for publication.
Thank you very much for your attention and courtesy.
Reviewer 2 Report
I accept new version. I have no comments.
Author Response
reviewer
I accept new version. I have no comments.
ANSWER
We thank the Reviewers for the considerable attention and the valuable comments that certainly helped us to improve the quality of the present paper.
Reviewer 3 Report
Dear Dr
The main limitation id a low number of studies in the meta-analysis and therefore I prefer to reject it.
Author Response
Referee's 3 report.
Dear Dr
The main limitation id a low number of studies in the meta-analysis and therefore I prefer to reject it.
Reply to Referee's 3 report:
We thank the Reviewer for the considerable attention and the valuable comments that certainly helped us to improve the quality of the present paper. We have improved the manuscript according to the Reviewers’ comments. A revision of the article has been carried out.
In particular, the number of studies included for the meta-analysis would represent a limit, but the limit is exceeded by the execution of the TSA in which it is deduced, even if slightly, how the data are statistically significant and present an adequate statistical power to cope with the type one and two errors (false negatives and false positives).
In addition, in Table 2, for greater clarity of the execution of the systematic review, additional studies are added in which the Di HR data on OS between high and low expression are highlighted, but these studies have not been included in meta-analyzes to avoid a risk of bias between studies, the reasons for exclusion have been extensively described. The inclusion of these data would lead to a probable spurious effect on the final effect.
Table 2. The data extracted for the four articles included in the meta-analysis and of the three excluded.; R (radiotherapy), OPMD (oral potentially malignant disorder) .
First autor, Data. |
country |
Study design |
number of patients |
Follow up max |
Tumor type\tumor site |
Cut-off |
miR |
HR miR-31 low and high expression (OS, PFS, CSS, DFS, RFS) |
Jakob (2019) [1] |
Germany |
RT |
36 |
60 months |
OSCC |
median |
miR‐21, miR‐29, miR‐31, miR‐99a, miR‐99b, miR‐100, miR‐143, miR‐155. |
OS:HR 3.69 (1.07-12.79) P = 0.028 RFS:HR 1.82(0.66-5.05) P = 2.4297 PFS: HR 2.31 (0.94-5.69) P = 0.05982
|
Wang (2018) [2] |
China |
RT |
118 |
60 months |
HNSCC |
median |
miR-31 |
OS: HR 3.31 (1.42-5.36) * P = 0.015 DFS: HR 3.86 (1.53-6.05) P=0.009 |
Qiang (2019) [3] |
China |
RT |
46 |
60 months |
21 HSCC , 25 LSCC |
median |
miR-31
|
OS: HR 1.38 (1.02-1.87) p = 0.036 |
Tu (2021) [4] |
Taiwan |
RT |
40 |
160 months |
OSCC |
median |
miR-31 |
OS: HR 1.68 (0.7747-3.6433) p = 0.189 |
Chen (2018)[5]1 |
China |
RT |
509 (307 con R) |
80 months |
HNSCC |
median |
miR-99a, miR-31 miR-410, miR-424, miR-495 |
OS R: HR 3.65, (2.46–8.16) P < 0.0001; OS: HR 1.81 (1.45-2.57) P < 0.0001 |
Gao (2013)[6]2 |
USA |
RT |
150 |
50 months |
OPSCC |
median |
miR-9, miR-223, miR-31, miR-18a, miR-155 |
OS HR 3.22 P = 0.0022 |
Hung (2016)[7]3 |
Taiwan |
Prospective |
46 |
28 months |
OPMD |
median |
miR-21, miR-31 |
PFS:HR 8.43 (1.04–68.03) P= 0.047 |
* There is probably an error in reporting the confidence interval values in the study by Wang et al. for values of p Value 0.015 and hazard ratio of 3.31 these confidence intervals seem not to be possible, in fact ReV Manager 5.4 reports for a confidence interval value of [1.42 5.36] and an HR of 2, 7. For the determination and implementation of the confidence intervals in the ReV manager 5.4 software we proceeded using the p value of 0.15 and HR 3.31;
1 Study not included in the meta-analysis because the HR value refers to the OS of HNSCC between high and low expression overall of 5 miR including miR-31; Furthermore, the data derive from the TGCA;
2 Study not included in the meta-analysis because the HR value refers to the OS of OPSCC between high and low expression overall of 5 miR (miR-9, miR-223, miR-31, miR-18a, miR-155);
3 The study was excluded because it reported data on PFS referring to potentially malignant precancerous conditions:
Thank you very much for your attention and courtesy.
References
- Jakob, M.; Mattes, L.M.; Küffer, S.; Unger, K.; Hess, J.; Bertlich, M.; Haubner, F.; Ihler, F.; Canis, M.; Weiss, B.G., et al. MicroRNA expression patterns in oral squamous cell carcinoma: hsa-mir-99b-3p and hsa-mir-100-5p as novel prognostic markers for oral cancer. Head and Neck 2019, 41, 3499-3515, doi:10.1002/hed.25866.
- Wang, L.-L.; Li, H.-X.; Yang, Y.-Y.; Su, Y.-L.; Lian, J.-S.; Li, T.; Xu, J.; Wang, X.-N.; Jin, N.; Liu, X.-F. MiR-31 is a potential biomarker for diagnosis of head and neck squamous cell carcinoma. International journal of clinical and experimental pathology 2018, 11, 4339-4345.
- Qiang, H.; Zhan, X.; Wang, W.; Cheng, Z.; Ma, S.; Jiang, C. A Study on the Correlations of the miR-31 Expression with the Pathogenesis and Prognosis of Head and Neck Squamous Cell Carcinoma. Cancer Biother Radiopharm 2019, 34, 189-195, doi:10.1089/cbr.2018.2621.
- Tu, H.-F.; Liu, C.-J.; Hung, W.-W.; Shieh, T.-M. Co-upregulation of miR-31 and its host gene lncRNA MIR31HG in oral squamous cell carcinoma. Journal of Dental Sciences 2021, https://doi.org/10.1016/j.jds.2021.11.006, doi:https://doi.org/10.1016/j.jds.2021.11.006.
- Chen, L.; Wen, Y.; Zhang, J.; Sun, W.; Lui, V.W.Y.; Wei, Y.; Chen, F.; Wen, W. Prediction of radiotherapy response with a 5-microRNA signature-based nomogram in head and neck squamous cell carcinoma. Cancer Med 2018, 7, 726-735, doi:10.1002/cam4.1369.
- Gao, G.; Gay, H.A.; Chernock, R.D.; Zhang, T.R.; Luo, J.; Thorstad, W.L.; Lewis Jr, J.S.; Wang, X. A microRNA expression signature for the prognosis of oropharyngeal squamous cell carcinoma. Cancer 2013, 119, 72-80, doi:https://doi.org/10.1002/cncr.27696.
- Hung, K.F.; Liu, C.J.; Chiu, P.C.; Lin, J.S.; Chang, K.W.; Shih, W.Y.; Kao, S.Y.; Tu, H.F. MicroRNA-31 upregulation predicts increased risk of progression of oral potentially malignant disorder. Oral Oncol 2016, 53, 42-47, doi:10.1016/j.oraloncology.2015.11.017.

Round 2
Reviewer 3 Report
DearAuthor (s)
There is no problem.
This manuscript is a resubmission of an earlier submission. The following is a list of the peer review reports and author responses from that submission.
Round 1
Reviewer 1 Report
This manuscript provides a meta-analysis of 4 studies (240 patients total) or miR-31 expression prognostic impact on HNSCC.
- (lines 41, and 226) Please change “Higgs Index” to “Higgins Index”.
- (line 145) Please correct the phrase “the k agreegment between the two reviewers”.
- Table 2) In the list of columns, please change “miR-21” to “miR-31”.
- (Figure 2) The 95% CIs for the Jakob and Wang studies do not match those given in Table 2. Please correct.
- (line 273) The phrase “Jakob et al and Wang et al report overlapping HR values for OS” is confusing. Does this mean overlapping with each other, asthis is true of the other studies as well?
- (line 279) Change “Kaplane Meier” to “Kaplan-Meier”.
- (line 280) The 95% CI limits can be limited to 2 significant digits.
- (line 284) Please correct errors in the sentence.
- (Study limitations) Consider mentioning that 75.6% of the weight of the meta-analysis finding derived from a single study (Wang et al).
- (lines 312-314) The conclusion could be stated more strongly, such as “In conclusion, the data from the meta-analysis suggest that elevated miR-31 expression is associated with poor prognosis in patients with HNSCC”. Something similar could be used in the abstract.
Reviewer 2 Report
in my opinion, 165-170 are redundant
215-318 this sentence is overused and does not add anything new, I recommend to change it.
Reviewer 3 Report
Dear Author (s)
- There are a lot of grammatical errors. Such as "Scopus PubMed, and the Cochrane databases", "concerning Overall survival", "through Funnel plot; If the meta", etc.
- Please add the references for PRISMA, REMARK, GRADE, bias, etc.
- Please add the name of the reviewers in the methods.
- "miR-31 AND HNSCC, Microrna AND HNSCC AND prognosis, miR-31 AND OSCC, miR-31 AND laryngeal cancer". You should select "head and neck cancer", oral cancer, oral carcinoma, oral squamous cell carcinoma, head and neck squamous cell carcinoma, pharyngeal cancer, etc. You lost several keywords in MeSH terms for searching. Therefore, your search is not systematic.
- Why did you select the Web of Science database?
- The number of studies is very low and therefore the conclusion is not logical and citable for future studies.
- Please add TSA.